# Guest-adaptive molecular sensing in a dynamic 3D covalent organic framework

Lei Wei[1,4], Tu Sun[1,2,4], Zhaolin Shi[1], Zezhao Xu[1], Wen Wen[3], Shan Jiang[1,2], Yingbo Zhao ®[1,2] ✉, Yanhang Ma ®[1,2] ✉ & Yue-Biao Zhang ®[1,2] ✉

Molecular recognition is an attractive approach to designing sensitive and selective sensors for volatile organic compounds (VOCs). Although organic macrocycles and cages have been well-developed for recognising organics by their adaptive pockets in liquids, porous solids for gas detection require a deliberate design balancing adaptability and robustness. Here we report a dynamic 3D covalent organic framework (dynaCOF) constructed from an environmentally sensitive fluorophore that can undergo concerted and adaptive structural transitions upon adsorption of gas and vapours. The COF is capable of rapid and reliable detection of various VOCs, even for non-polar hydrocarbon gas under humid conditions. The adaptive guest inclusion amplifies the host-guest interactions and facilitates the differentiation of organic vapours by their polarity and sizes/shapes, and the covalently linked 3D interwoven networks ensure the robustness and coherency of the materials. The present result paves the way for multiplex fluorescence sensing of various VOCs with molecular-specific responses.

It is a longstanding challenge for conventional electronic gas sensors to simultaneously identify the type and determine the concentration of an organic gas/vapour from various volatile organic compounds (VOCs)[1–3]. Fluorescence sensing can be a viable approach to tackle this challenge[4–7], which requires molecular recognition to enable the weak and non-specific interactions between VOC molecules and solid-state fluorescence probes to give deterministic fluorescence responses. Molecular recognition[4,8,9] has been widely exercised with organic macrocycles and cages in solutions and liquids, where complementary shape adaptivity and well-defined supramolecular interactions can give specific chemical responses. In contrast, the solid-state dynamic adaptivity for gaseous molecular recognition remains largely undeveloped[10–15].

Soft porous crystals (SPCs) have been known to show cooperative phenomena of host-guest adaptability, guest-guest arrangement, and framework switchability accompanied by collective structural transitions, which can give dynamic and deterministic responses upon guest inclusions[16–25]. However, for most of these frameworks, their conformational changes rely on geometry changes and/or dissociation/reformation of weak bonds[26–33], imposing concerns about the stability and durability in practical scenarios for gas sensing. Thus, the pursuit of robust and dynamic framework materials has been focused on the 3D covalent organic frameworks (COFs)[34–38]. Recently, we and others have uncovered the dynamic responses and determined the structural transformation[39–48] of dynamic 3D COFs (dynaCOF) based on the interwoven diamond networks with revolving imine bonds[41–43]. We anticipate that installing fluorophores on dynaCOFs might enable the adaptive inclusion of organic gas/vapour molecules to produce and amplify deterministic fluorescence responses (Fig. 1).

In this contribution, we report the design, synthesis, and characterisation of fluorescent and dynamic 3D COFs, dynaCOF-330, incorporating an environmentally sensitive anthracene-derivative fluorophore[49] on the framework scaffold. The material undergoes different structural transformations when exposed to various organic

[1]School of Physical Science and Technology, ShanghaiTech University, Shanghai 201210, China. [2]Shanghai Key Laboratory of High-Resolution Electron Microscopy, ShanghaiTech University, Shanghai 201210, China. [3]Shanghai Synchrotron Radiation Facility, Shanghai Advanced Research Institute, Chinese Academy of Sciences, Shanghai 201210, China. [4]These authors contributed equally: Lei Wei, Tu Sun. ✉e-mail: zhaoyb2@shanghaitech.edu.cn; mayh2@shanghaitech.edu.cn; zhangyb@shanghaitech.edu.cn

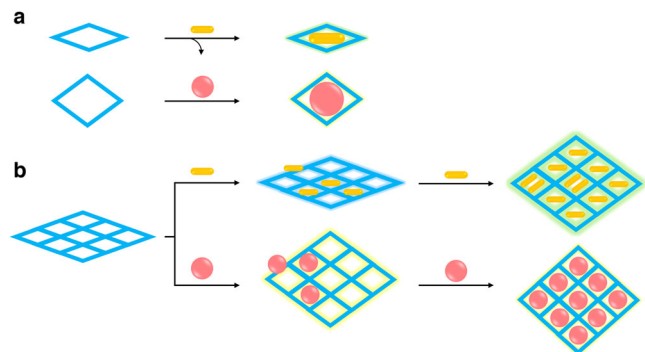

**Fig. 1 | The working principle of guest-adaptive molecular sensing for dyna-COFs. a** Traditional molecular recognition of the sizes and shapes of guest molecules via the self-adaptivity of supramolecular macrocycles and cages for molecular sensing in liquids (solutions). **b** Dynamic and functional confinement spaces for the docking of guest molecules and the adaptive inclusion of more guests to output molecular-specific fluorescent responses varied in intensity and colours for multiplex molecular sensing of gas/vapours.

vapours, which are accompanied by variations in emission wavelength and fluorescence intensity. The correlation between fluorescence spectra and the type and partial pressure of the vapour enables reproducible quantification of a wide variety of organic vapours. The molecular insights of the fluorescence responses are uncovered by combining 3D electron diffraction (ED), synchrotron powder X-ray diffraction (PXRD), vapour adsorption isotherms, in-situ fluorescence spectroscopy, ex-situ PXRD analyses, and molecular simulation. The docking of guest molecules in the pores restricts the vibration of anthracene to enhance the fluorescence intensity; and interferes with the donor-acceptor electronic conjugated system to modulate the emission wavelength. The extent and direction of the fluorescence intensity and wavelength changes strongly depend on the framework's flexibility due to the adaptive inclusion, which enables the differentiation and quantification of a wide variety of organic vapours. The dynaCOF-330 shows rapid responses (as fast as 1 s), high sensitivity (1% for *n*-butane), high stability (>500 cycles), and tolerance to humidity (53% RH), which hold great promise for practical VOC detection aided by the adaptive guest inclusion.

## Results and discussions

### Design and synthesis of fluorescent dynaCOF

The design of the dynaCOF relies on the revolving of imine bonds for adaptive guest inclusion. The environmentally sensitive probes are installed as struts for fluorescence sensing of guest molecules (Fig. 2a). The 4,4′-(2,3,6,7-tetramethoxyanthracene-9,10-diyl)dibenzaldehyde (AnDA), featuring an acceptor-donor-acceptor (A-D-A) π-conjugated system, has been studied by us to have an ultrafast fluorescence switch-on and colour-tuned dynamics[49]. The dynaCOF-330 is prepared through the ventilation-vial synthetic protocol[42] via the imine condensation of AnDA with 4,4′,4″,4‴-methanetetrayltetraaniline (TAM) in 1,4-dioxane to form a 3D interwoven diamondoid (**dia**) network (Fig. 2b, Supplementary Fig. 1-4), which is further activated under vacuum and heating. The formation of imine bonds and the elimination of aldehyde starting materials has been confirmed by Fourier-transformed infrared spectroscopy (FT-IR, Supplementary Fig. 5, solid-state nuclear magnetic resonance spectroscopy (ssNMR, Supplementary Fig. 9), and high-resolution X-ray photoelectron spectroscopy (XPS, Supplementary Fig. 7). The morphology and crystallinity of dynaCOF-330 were examined by scanning electron microscopy (Supplementary Fig. 8) and PXRD patterns (Supplementary Fig. 6). DynaCOF-330 can stabilize up to 430 °C, characterised by thermal gravimetric analysis (Supplementary Fig. 10) and stabilize in an alkali solution (Supplementary Fig. 13-14). The dynamic nature of dynaCOF-

330 is visualised by the stepwise and hysteretic $CO_2$ adsorption isotherm at 195 K (Supplementary Fig. 11) to approach a pore volume of 0.73 cm³/g, which is coincident with the change of PXRD pattern of the activated and re-solvated sample (Supplementary Fig. 12).

### Structural determination by 3D ED and synchrotron PXRD analyses

Given the high crystallinity and well-defined morphology, the crystal structure of the activated dynaCOF-330 was determined by combining single-crystal 3D electron diffraction (3D ED, Supplementary Section 2) and synchrotron PXRD (Supplementary Section 3). The microcrystals of dynaCOF-330 represent a well-defined tetragonal prism shape (Fig. 2c) with a narrow distribution of crystal sizes[44] (1.7-1.9 μm, Supplementary Fig. 8). The activated dynaCOF-330 crystallises in the orthorhombic crystal system with a space group of *Pnn*2 (No. 34), and the unit cell parameters of $a = 16.838(1)$ Å, $b = 29.116(1)$ Å, and $c = 7.639$ (1) Å, of which the structure can be determined by using direct-space method against the resolution-limited (~2.0 Å) 3D ED data (Fig. 2d; Supplementary Figs. 15-16)[45]. Although the dynamic nature of porous organic crystals led to low resolution under the electron beam, the high-resolution synchrotron PXRD pattern of the activated dynaCOF-330 (Fig. 2e) allows the Rietveld refinement ($R_{wp} = 5.9\%$ and $R_p = 1.2\%$) against the structural model constructed based on the 3D ED results (Supplementary Figs. 17, 20, and Supplementary Data 1). Indeed, dyanCOF-330 adapts the 3D diamondoid (**dia**) networks interweaving along the *c* axis to have 10-fold catenations (Fig. 2g). The functional anthracene motifs stack into the sliding column to have an intermolecular separation of 7.6 Å (Fig. 2h), preventing the photocyclisation reaction[50,51] for stable fluorescence sensing. Channels along the *c* axis are squeezed by the bulking organic linker due to the mobile imine linkages to minimise the porosity as low as 13.3%. This is crucial for having precise molecular recognition and maximal host-guest interactions.

### Proportional fluorescence responses upon adaptive inclusion of acetone vapour

The fluorescence turn-on effect of dynaCOF-330 is observed by the naked eye upon *n*-butane gas and organic vapour uptake. To facilitate the study of the mechanism of fluorescence response, acetone and 1,4-dioxane vapours have been selected for representative organic vapours, as their different structural transformation and fluorescence response, and vapour adsorption isotherms, in-situ fluorescence spectroscopy (Supplementary Section 4), and ex-situ synchrotron PXRD have been performed (Supplementary Section 3). The vapour adsorption isotherm of acetone at 298 K (Fig. 3a) exhibits a stepwise uptake, and the fluorescence turn-on can be observed by the naked eye at 0.041kPa. The fluorescence intensity increases monotonically in the entire pressure range. The emission wavelength ($\lambda_{em}$) redshifts with increasing acetone pressure from 490 to 518 nm (Fig. 3b). Remarkably, the fluorescence responses can be correlated to the acetone uptake, which can be divided into four pressure ranges (Fig. 3c; Supplementary Fig. 35): 0-0.15 kPa (stage I, blue), both the acetone uptake and the fluorescence intensity increase with pressure; 0.15-4.8 kPa (stage II, green), the fluorescence redshift to 505 nm and the intensify rapidly increase along with the steep uptake of acetone; 4.8-24 kPa (state III, yellow), the fluorescence further redshift to 518 nm and both the fluorescence intensity and acetone uptake mostly plateaus; 24-29 kPa (state IV, red), the fluorescence intensity shows a steeper increase along with the uptake of acetone. The fluorescence intensity increases by eight times at 4.76 kPa. The highest intensity is observed at 29 kPa with a 10-folded increase in fluorescent intensity. Steady sensing of acetone vapours is illustrated in situ by the vacuum swing of acetone vapour with the pressure of 28.5 kPa, showing the robustness of dynaCOF upon adaptive inclusion (inset, Fig. 3c).

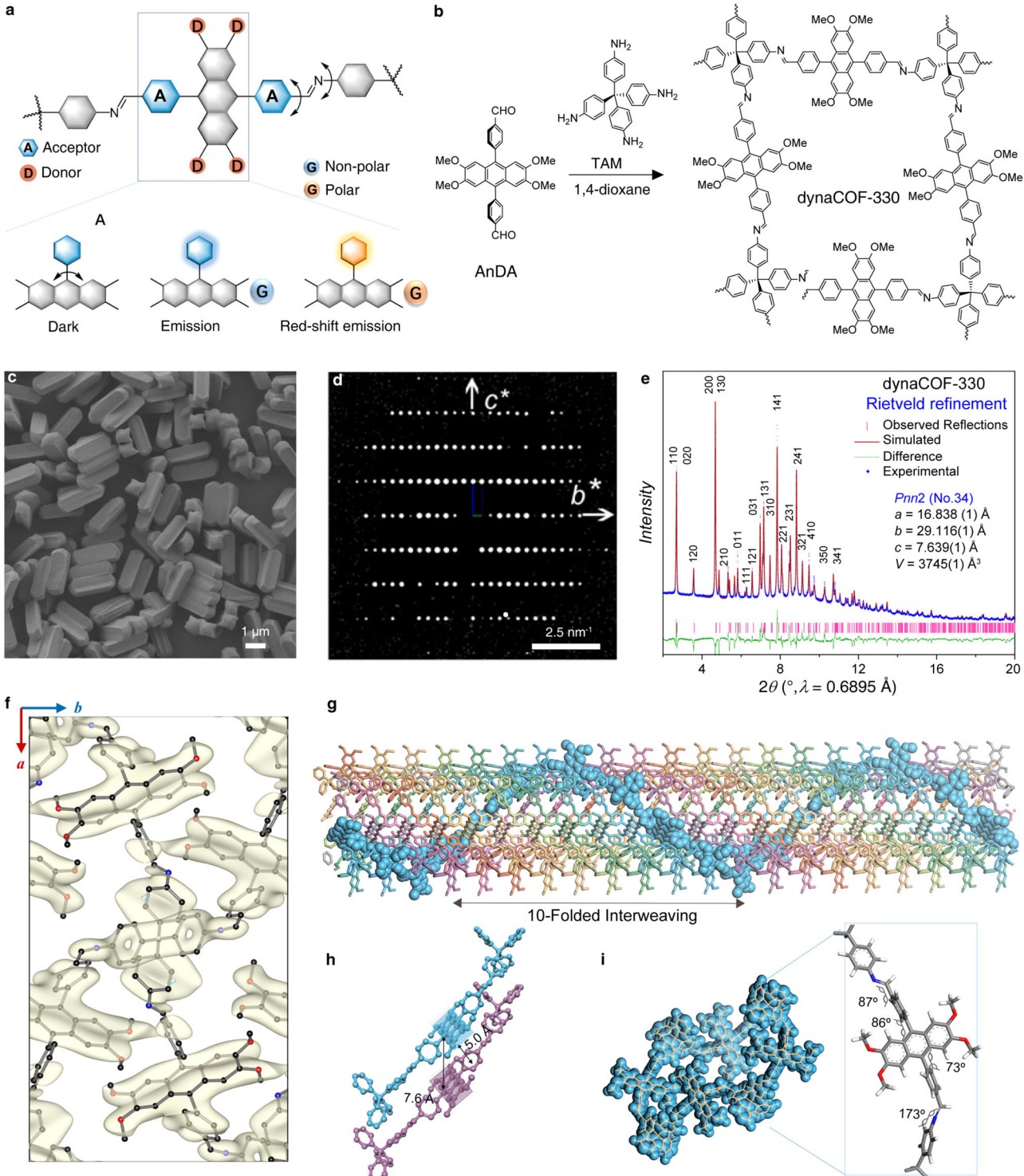

**Fig. 2 | Design, preparation, and the crystal structure determination of dynaCOF-330. a** Mechanism of fluorescence turn-on by hindering the intermolecular flapping of fluorophore and the fluorescence colour tuned by chemical environments. **b** Solvothermal synthesis of the crystalline sample by imine condensation of AnDA with TAM in 1,4-dioxane. **c** Morphology of the microcrystals showing well-defined tetragonal prismatic shapes and uniform sizes. **d** Single-crystal 3D ED data reconstructed in reciprocal space projected along the [100] direction for determining lattice constants and space group symmetry. **e** Rietveld refinement of crystal structure obtained from the 3D ED data against the

synchrotron PXRD pattern of the guest-free sample. **f** Comparison of the refined crystal structure with the electrostatic potential map generated from the 3D ED data. **g** Interweaving structure of the characteristic $4_1$ helix chains in the 10-fold catenation of the diamondoid networks. **h** The stacking fashion of fluorophores with a sliding distance of 7.6 Å and an interlayer distance of 5.0 Å for photo-stabilization of anthracene motifs. **i** Space-filling view of channels projected along the $c$ axis with slim pore opening and representative geometries for the conformation of the organic linkers.

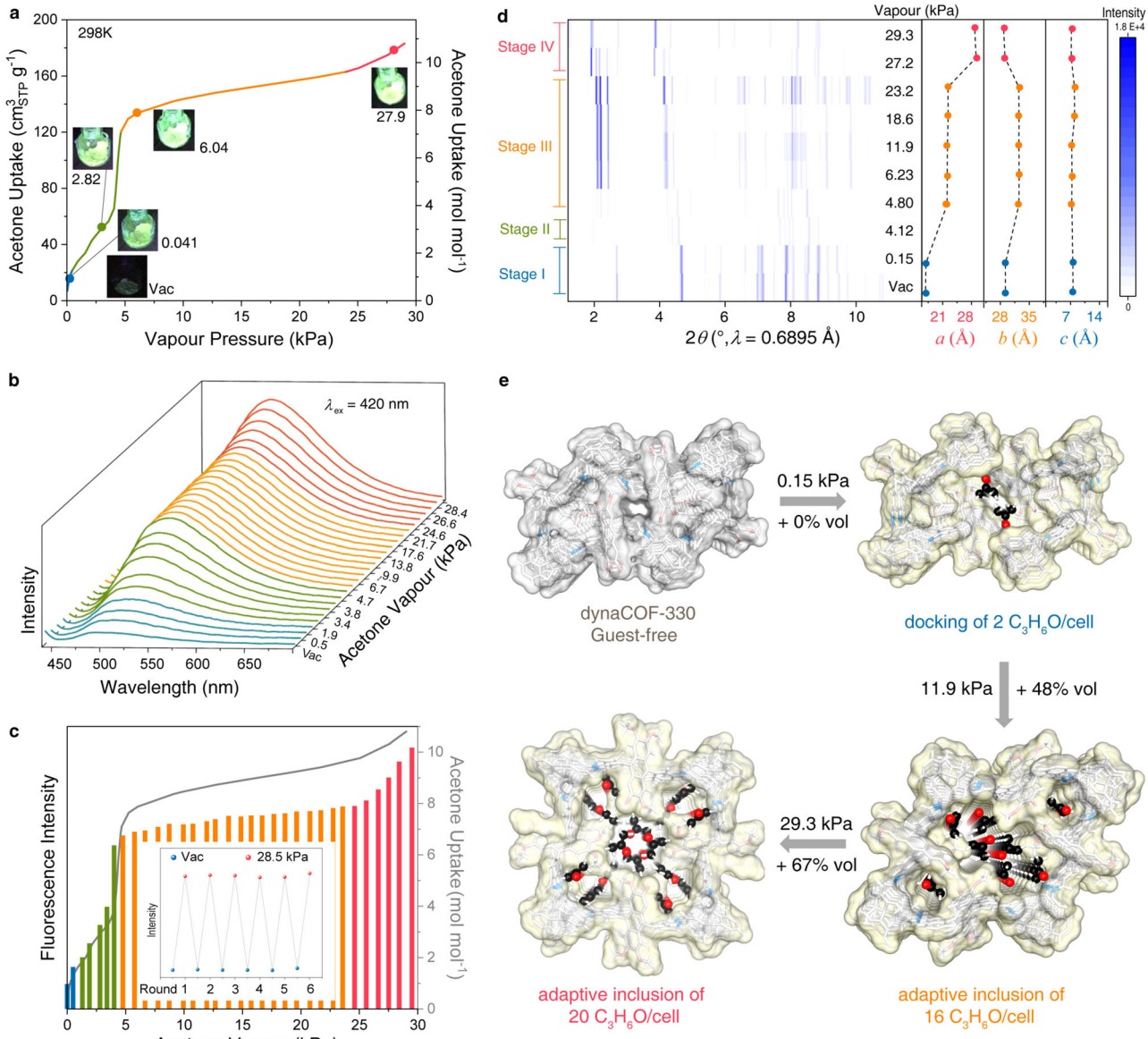

**Fig. 3 | Proportional fluorescence response and structural transition of dyanCOF-330 upon docking and adaptive inclusion of acetone vapour.** **a** Optical images of dynaCOF-330 under 365-nm UV lights showing the fluorescence turn-on and colour change for representative equilibrium pressures and uptakes of acetone vapour at 298 K. **b** Quantitative in-situ fluorescence spectroscopy of dynaCOF-330 showing maxima redshift of emission wavelength. **c** Correlation of fluorescence emission intensity with acetone vapour uptake with an inset for the steady acetone vapour sensing through adaptive inclusion. **d** Lattice parameters refined from ex-situ PXRD of dynaCOF-330 at representative equilibrium pressures of acetone vapour. **e** Evolution of crystal structures of dynaCOF-330 upon docking and adaptive inclusion of various numbers of acetone molecules per cell. (Colour codes: C, black and grey; N, blue; O, red; H, white. Connolly pore surfaces are shown in grey and yellow.).

The synchrotron ex-situ PXRD patterns also show the dynaCOF-330 appears to have four distinct crystalline phases after the adaptive inclusion of different amounts of acetone molecules (Fig. 3d; Supplementary Fig. 21; Supplementary Table 1). The framework structures of dynaCOF-330 and locations of acetone molecules at different stages were all determined by refinement against synchrotron PXRD patterns. In stage I (0-0.15 kPa), no noticeable change is observed in the PXRD patterns of dynaCOF-330. At 0.15 kPa, about two acetone molecules are adsorbed per unit cell, and the docking site of acetone molecules is elucidated by molecular simulation in combination with PXRD refinement (Supplementary Fig. 22). The fluorescence turn-on effect can be attributed to the restrained vibration of the anthracene core of the fluorescence probe. In stage II (0.15-4.8 kPa), the structure shows significant disordering, and lattice parameters cannot be determined. However, extrapolating from the lattice parameter of the previous and

next adsorption stage and considering a large amount of acetone is being adsorbed at this stage, the framework should also be expanding in this pressure range. The framework conformational change, combined with acetone inclusion, alters the polarity and polarizability of the pore environment and results in a red shift in the fluorescence spectra. In stage III (4.8-27 kPa), beyond 4.8 kPa, the framework enters a crystalline intermediate state, and the lattice parameter remains relatively constant, consistent with the plateaued fluorescence intensity and acetone uptake. The intermediate state has rhomboid channels, unit cell parameters of $a = 22.18$ Å, $b = 7.32$ Å, $c = 32.50$ Å, $\beta = 92.92°$, and space group of $P2/n$, which is determined by PXRD Rietveld refinement with $R_{wp} = 4.5\%$ and $R_p = 1.6\%$ (Fig. 3e; Supplementary Fig. 23; Supplementary Data 2). The framework conformation changes to accommodate the acetone molecules have led to further fluorescence redshift than in the previous stage. In stage IV (>27 kPa),

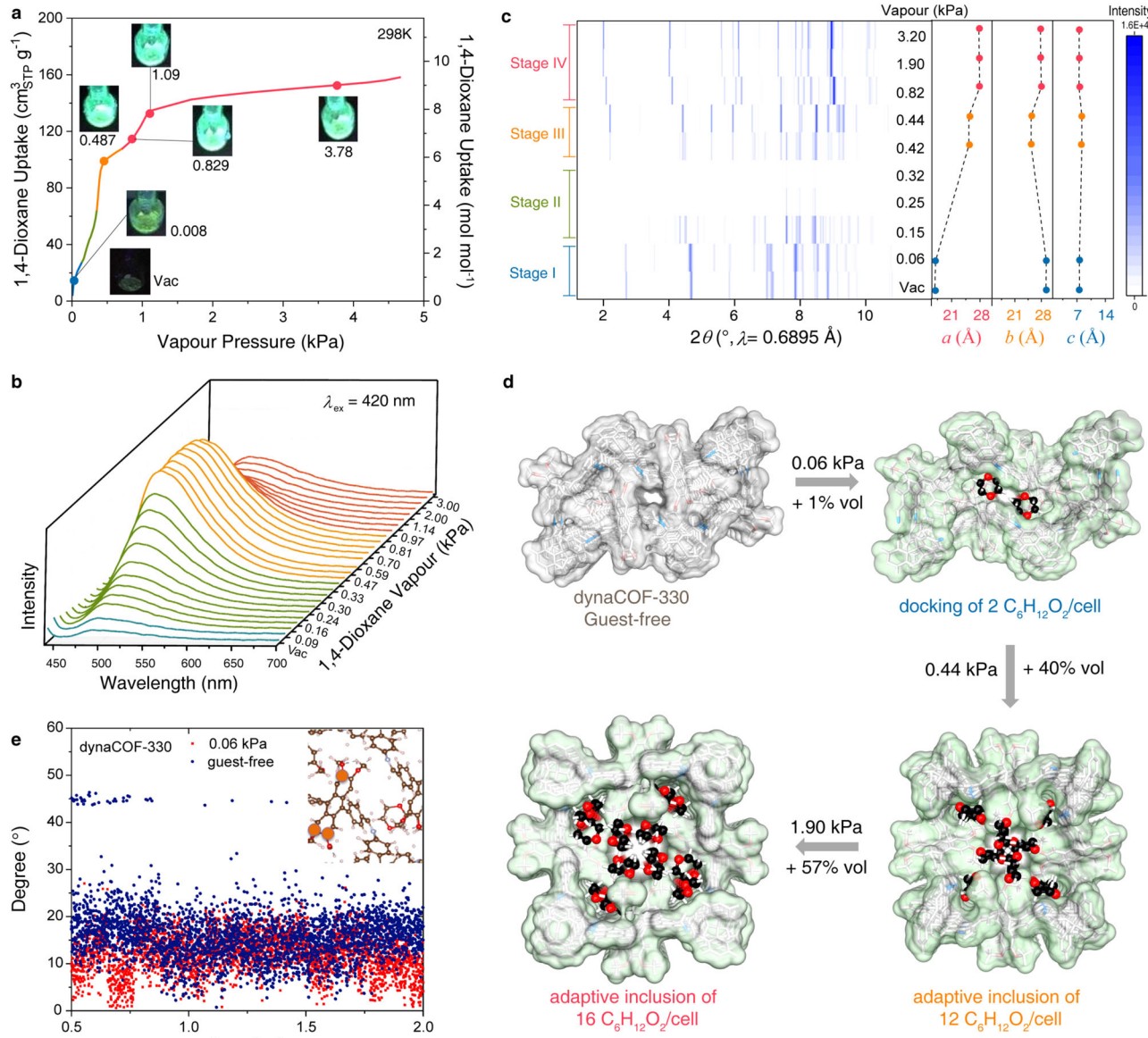

**Fig. 4 | Anomalous fluorescence response and structural transition of dynaCOF-330 upon adaptive docking and inclusion of 1,4-dioxane vapour.** **a** Optical images of dynaCOF-330 under 365-nm UV lights showing the fluorescence turn-on and intensity changes for representative equilibrium pressures and uptake of 1,4-dioxane vapour at 298 K. **b** Quantitative in-situ fluorescence spectroscopy of dynaCOF-330 showing constant maxima of emission wavelength. **c** Lattice parameters refined from ex-situ PXRD of dynaCOF-330 at representative equilibrium pressures of 1,4-dioxane vapour. **d** Evolution of crystal structures of dynaCOF-330 upon docking and adaptive inclusion of various numbers of 1,4-dioxane molecules per cell. (Colour codes: C, black and grey; N, blue; O, red; H, white. Connolly pore surfaces are shown in grey and green.). **e** The variation of dihedral angle changes of the anthracene plane between the selected time and the initial one, and the insert figure for the plane defined by the three carbon atoms marked by the orange circle.

---

the framework transits into the expanded state with a tetragonal channel and different symmetry (Supplementary Fig. 24; Supplementary Data 3), corresponding to a kink in acetone adsorption isotherm and intensifying the fluorescence emission.

## Anomalous fluorescence responses and guest-guest rearrangement for 1,4-dioxane

The guest-dependence of fluorescence response in dynaCOF-330 is observed in the study of adaptive inclusion of 1,4-dioxane with similar polarity, but larger size than acetone (Fig. 4). The vapour adsorption isotherms of 1,4-dioxane at 298 K (Fig. 4a) also show stepwise uptakes, fluorescence turn at 0.008 kPa, and significant intensity enhancement can be observed at 0.487 kPa. However, the in-situ fluorescence spectroscopy shows the constant emission wavelength ($\lambda_{max}$ = 490 nm, Fig. 4b; Supplementary Fig. 36). Based on the slope of

intensity change, the 1,4-dioxane uptakes can also be divided into four stages (Supplementary Fig. 37), which represent intensity increment in stages I (0-0.13 kPa), II (0.13-0.37 kPa), and III (0.37-0.81 kPa), but intensity decrement in stage IV (0.81-4.5 kPa). The fluorescence sensing of 1,4-dioxane is reproducible by the vacuum-pressure swing of 1,4-dioxane pressure of 0.5 kPa and 3.3 kPa (Supplementary Figs. 37-38).

The representative ex-situ PXRD patterns of dynaCOF-330 also show four distinct phases upon the inclusion of a varied number of 1,4-dioxane molecules (Fig. 4c; Supplementary Fig. 26). At 0.06 kPa of 1,4-dioxane vapour pressure, vapour adsorption isotherm shows that, on average, two 1,4-dioxane molecules are adsorbed in each unit cell (Supplementary Data 4;). The framework maintains the same conformation as the empty state. From Rietveld refinement analysis ($R_{wp}$ = 4.3% and $R_p$ = 1.9%, Supplementary Fig. 27), the two 1,4-dioxane

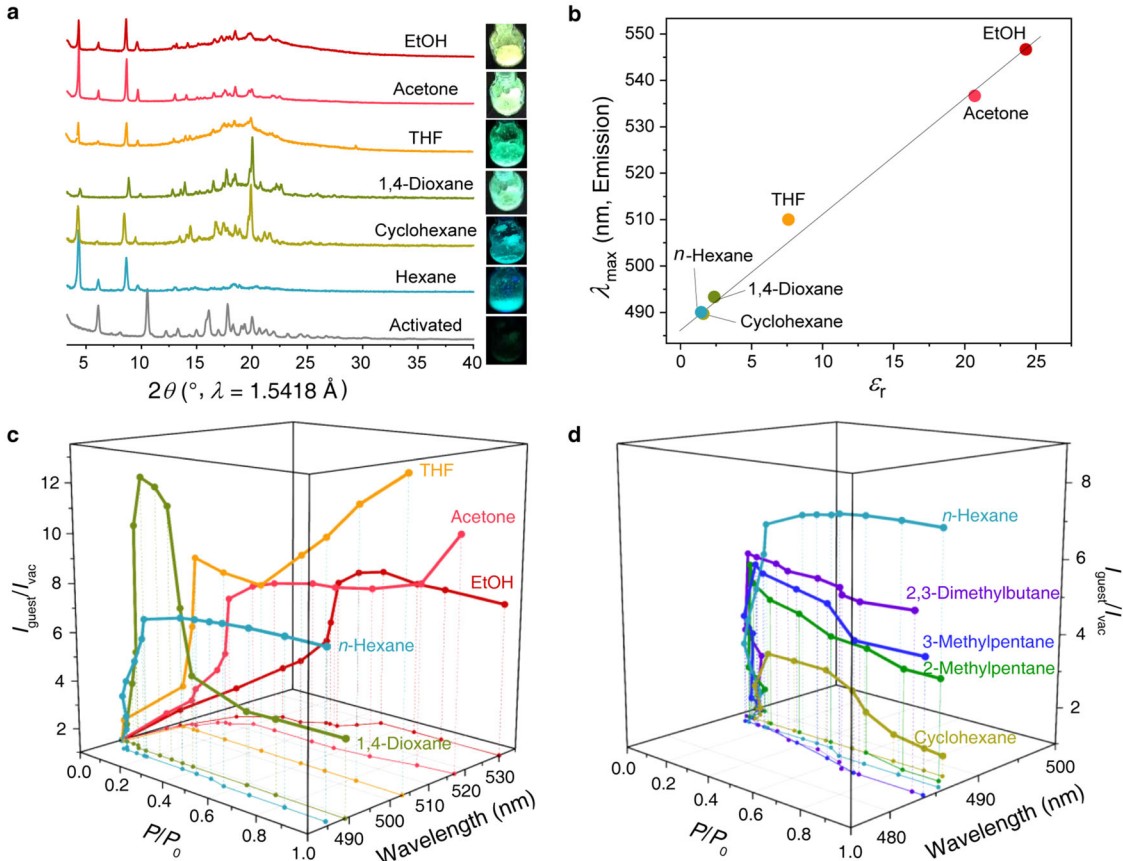

**Fig. 5 | Molecular sensing is endowed by the universality of adaptive guest inclusion of dynaCOF-330. a** Fluorescence responses upon adaptive inclusion of various saturated organic vapours indicated by the changes of PXRD patterns. **b** Guest polarity ($\varepsilon_r$, dielectric constant at 25 °C) dependence of emission maximum wavelengths for dynaCOF-330 upon docking two organic molecules per unit cell. **c** Diverse fluorescence responses of dynaCOF-330 to various organic vapours with distinct polarity. **d** Distinguishable fluorescence responses of dynaCOF-330 to C6 hydrocarbon isomers vapours with similar polarity.

molecules are located near the anthracene cores (Fig. 4d, right column), which hinders the intra-molecular flapping of the fluorescence probe, enhancing the emission intensity. The restriction of anthracene core by adsorbed 1,4-dioxane molecules is also supported by molecular dynamic calculations, which show smaller vibration amplitude for the anthracene cores (Fig. 4e). From 0.06 to 0.44 kPa, the framework first goes through a non-crystalline state, then enters an intermediate crystalline phase at 0.42 kPa. The fluorescence increases monotonically in this range, reaching a maximum of 0.44 kPa. From 0.44 to 0.8 kPa, the framework is in the intermediate state with fluorescence and lattice constant relatively unchanged (Fig. 4c; Supplementary Table 2). At 0.82 kPa, the material enters the expanded phase, with the channel expanding from 16×16 Å to 19×19 Å (Supplementary Fig. 31, Supplementary Data 5 and Data 6), and the unit cell parameters changes from $a = b = 24.96$ Å; $c = 8.00$ Å (obtained from Rietveld refinement with $R_{wp} = 7.5\%$ and $R_p = 5.2\%$, Supplementary Fig. 28) to $a = b = 27.84$ Å; $c = 7.26$ Å ($R_{wp} = 7.8\%$ and $R_p = 5.4\%$, Supplementary Fig. 29). In the expanded phase, the 1,4-dioxane molecules redistribute in the pore, accompanied by decreasing fluorescence emission intensity. Such a decrease is due to the expansion of the pore centre, which reduces the interactions between 1,4-dioxane and the anthracene core (Supplementary Fig. 32).

### Multiplex fluorescence responses for various organic vapours
The molecular-level insight gained in the study of 1,4-dioxane and acetone uptake as model systems can be generalised to a library of organic vapours, showing fluorescence turn-on and adaptive framework structural changes (Fig. 5a). The emission wavelength of

dynaCOF-330 also shifts with the increased polarity of the adsorbed molecules, providing selectivity and specificity to the fluorescence gas sensor (Fig. 5b). For different vapours, the adaptive feature of dynaCOF-330 gives rise to different fluorescence response profiles (Fig. 5c; Supplementary Figs. 40-41). Remarkably, even isomers of hexane give different fluorescence responses (Fig. 5d; Supplementary Fig. 42). These complex pressure-dependent fluorescence responses result from the different host-guest and guest-guest interactions in the adaptive molecular pockets.

### Fluorescence sensing of hydrocarbon gas under dry and humid conditions
The fluorescence turn-on effect of dynaCOF-330 is also observed for the adaptive inclusion of hydrocarbon gas. The *n*-butane adsorption isotherm of dynaCOF-330 is also stepwise and hysteresis (Fig. 6a) with the uptake of 150 cm³g⁻¹ and pore volume of 0.76 cm³g⁻¹ at 273 K and 1 bar, indicating the ability for adaptive inclusion. The fluorescence turn-on effect is observed by naked eyes during *n*-butane adsorption at 298 K with the irradiation of UV light (Fig. 6b), which glows blue light at 3 kPa and becomes shining at 40 kPa upon adaptive inclusion of 90 cm³g⁻¹ of *n*-butane. And we studied the sensitivity and reliability of fluorescence sensing of dynaCOF-330 by *n*-butane as an example since its weak interaction often leads to insensitive and slower response time. Generally, sensing conditions are always at room temperature and pressure, so a home-build setup (Supplementary Fig. 45) was connected to fluorescence spectroscopy to examine the performance of dynaCOF-330 for *n*-butane fluorescence sensing. With dry *n*-butane, 2% by volume is adequate to induce a fluorescence intensity change of

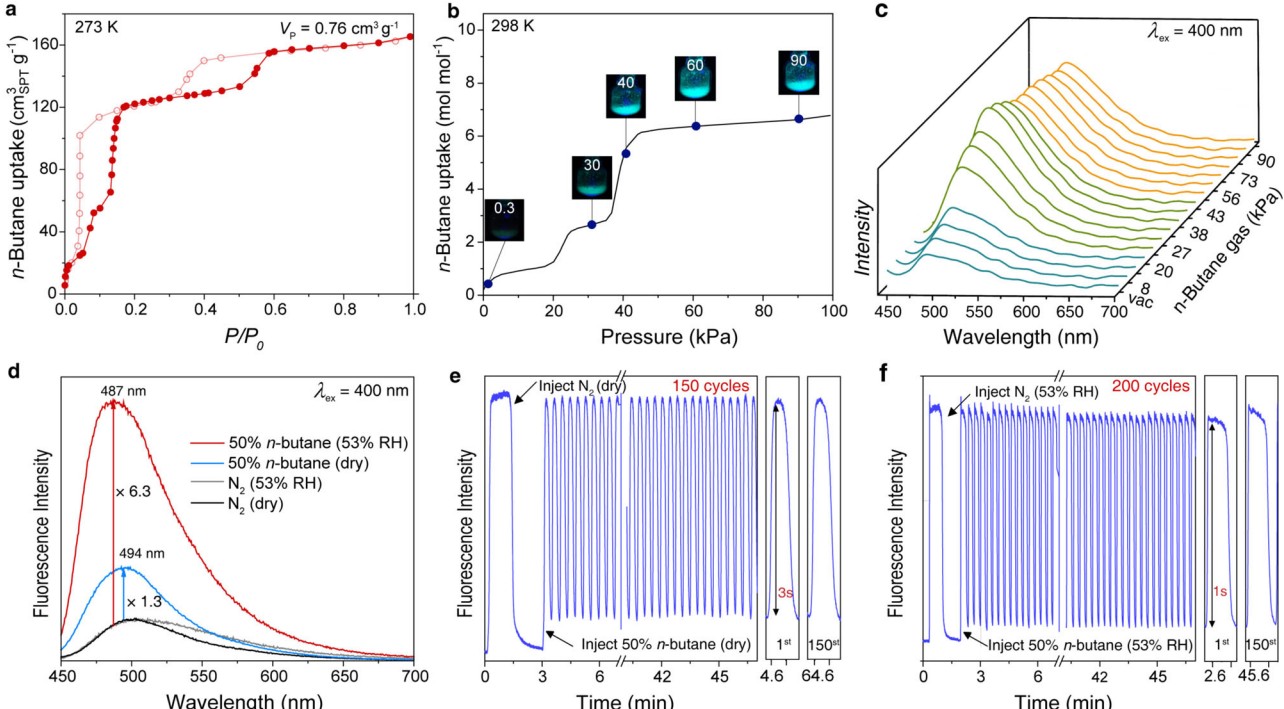

**Fig. 6 | The sensing performance of dynaCOF-330 for *n*-butane. a** Stepwise and hysteresis gas adsorption isotherm of *n*-butane at 273 K revealing the adaptive guest inclusion. **b** Representative optical images of dynaCOF-330 under 365-nm UV lights showing the fluorescence intensity enhancement upon *n*-butane uptakes at 298 K. **c** Quantitative in-situ fluorescence spectroscopy at representative equilibrium pressures during the *n*-butane adsorption at 298 K. **d** Significant fluorescence blueshift and boosted fluorescence intensity of dynaCOF-330 upon injection of 50% *n*-butane under dry and humid condition (53%RH). The cyclic fluorescence sensing of 50% *n*-butane under dry (**e**) and humid conditions (**f**, 53% RH) illustrates the robustness of dynaCOF-330 for rapid and steady sensing of non-polar organic gas under humid conditions.

10% (Supplementary Fig. 46), and this is the most sensitive fluorescence response reported. Humidity often interferes with gas detection, and we also examined other air components that almost have no fluorescence response (Supplementary Fig. 43). Surprisingly, the dynaCOF-330 becomes two times more sensitive to *n*-butane under 53% RH (Supplementary Fig. 47). Upon exposure to 50% *n*-butane, the fluorescence turn-on effect is boosted almost five times in comparison with the dry conditions (Fig. 6d), which can be attributed to the water adsorption in the pore (Supplementary Fig. 48) to reduce further the pore size for the adaptive inclusion of *n*-butane. The fluorescence sensing for *n*-butane is rapid and steady for at least 150 and 200 cycles under both dry and humid conditions (Fig. 6e, f; Supplementary Fig. 49), respectively. A fast response was observed with 3 s and 1 s for the fluorescent intensity increase upon switching from a dry $N_2$ and humid $N_2$ steam to n-butane, respectively.

In conclusion, we illustrated the adaptive guest inclusion in dynaCOF-330 for fluorescence sensing of various organic molecules in the gas phase. The resilience of the covalent bond endows the dynaCOF with steady and humidity-tolerant sensing in practical conditions. The concerted structural transition and the environmentally sensitive probe yield sensitive, specific and quantitative fluorescence responses through adaptive guest inclusion to maximise the host-guest interactions.

Specifically, dynaCOFs possess two-fold advantages suitable for molecular sensing in the gas/vapour phases: (a) the dynamic and porous nature of the framework would allow for facile adsorption of organic vapours in the gas phase that can give adaptive framework structural changes with high sensitivity and selectivity; (b) the covalently bonded framework has high robustness that gives coherent structural changes of the entire framework during guest inclusion without creating local defects, which ensures quantitative fluorescence response reflecting the extent of guest inclusion. Thus, the

adaptive structural changes of the dynaCOF enable the weak interactions associated with the physisorption of vapours to collectively and coherently modulate the chemical environment of the fluorophore installed on the framework to give characteristic fluorescence responses.

## Methods
### Synthesis of dynaCOF-330
AnDA (60 mg, 0.119 mmol) was dispersed in 2.5 mL of 1,4-dioxane, and then 0.5 mL of aqueous acetic acid (6 M) and 150 $\mu$L of aniline were added to a 20 mL vial. After that, sonication is needed to make the solid dispersed, and then the solution of TAM (25 mg, 0.066 mmol) dissolved in 2.5 mL of 1,4-dioxane was added. The reaction was heated at 80 °C for three days; the shallow yellow solid was isolated by a centrifuge, separated, and washed with 1,4-dioxane three times, then evacuated at 100 °C for 10 hours to get dynaCOF-330 (45 mg, 62% yield).

### In-situ fluorescence experiment of single component vapour
A in-house sample cell was prepared, as shown in Supplementary Fig. 34. dynaCOF-330 (about 40 mg) was filled in one side of the cell, avoiding scattering during the fluorescent experiment. Then the sample-filled cell was placed in a quartz adsorption cell with an inner diameter of 9.0 mm. Before fluorescence spectrum measurement, the piece was treated in a vacuum at 100 °C for 24 hours to ensure the guest was entirely removed. No thermostat was used during organic vapour uptake, but the temperature was monitored with a thermometer during the measurement showing the temperature fluctuated less than 1 °C. Solvents used for vapour adsorption are degassed at least five times before isotherm collection and gas/vapour dosing. Organic vapour dosing and pressure controlling were performed by BELSORP-max, which was connected to a photoluminescence

spectrometer to realise in-situ fluorescence experiments. An equilibration time of 1800 – 3600 s at each equilibrium pressure was adopted to ensure the equilibrium conditions of 1% pressure change within 300 s.

### In-situ fluorescence experiment for *n*-butane recycling responses

A homemade relative humidity and gas partial pressure controller was prepared, as shown in Supplementary Fig. 45. Two mass flow controllers (MFCs) with different controlling ranges (100 sccm, and 5 sccm for low partial pressure of less than 5%) were connected to the *n*-$C_4H_{10}$ cylinder, and two MFCs were connected to the $N_2$ cylinder as a purge gas to activate the sample or balanced gas mixed with *n*-$C_4H_{10}$. The $N_2$ and *n*-$C_4H_{10}$ mixture was then passed through a water bottle with saturated $MgCl_2$ to form the working gas with 53% RH. A tee valve was used to switch the wet working gas to dry gas. All the stainless-steel valves and joints were purchased from Shanghai X-tec Fluid Technology Co, Ltd, and the MFCs were purchased from Alicat Scientific (A Halma company). The sample was prepared as COF film by dropping a slurry of dynaCOF-330 (10 mg dispersed in 3 ml acetone) in the washed glass slide (1 cm × 3 cm) to measure its fluorescence response for *n*-butane gas. Before measurement, the COF film was dried under vacuum at room temperature for 5 hours to ensure the guest was entirely removed.

### Ex-situ synchrotron PXRD experiments

Vapor-loading PXRD experiments were measured at BL14B1 of the Shanghai Synchrotron Radiation Facility (SSRF). dynaCOF-330 was filled in the borosilicate glass capillary with an outer diameter of 0.8 mm and a thickness of 0.01 mm, continuously spinning during the experiments to improve data statistics. The monochromatic X-ray beam with an energy of 18 keV ($\lambda = 0.6895\,\text{Å}$) and a beam size of 180 μm (width) × 200 μm (height) was adopted. The Mython 1 K linear detector was adopted for high-resolution PXRD data acquisition in Debye-Scherrer mode. The wavelength of the X-ray was calibrated using the $LaB_6$ standard from NIST(660b). The transmission in PXRD patterns was collected at room temperature. The sample was filled in the capillary about 2 cm in length, and before vapour dosing vacuum overnight was needed. At each selected pressure, at least equilibrium for 4 hours to ensure the samples are in the same state.

### 3D ED data collection for the 1,4-dioxane vapour included dynaCOF-330

We prepared the guest-contained sample by fumigation. At first, the synthesised dynaCOF-330 sample was fully activated to ensure no guest molecules existed in the channel of dynaCOF-330 crystals. The activated sample was then dispersed into the 1,4-dioxane solvent to absorb 1,4-dioxane molecules fully. One droplet of suspension was transferred onto a carbon film-supported TEM grid. The gird was further air-dried for 30 minutes to thoroughly remove 1,4-dioxane solvent covered on the surface of dynaCOF-330 crystals. The grid with the sample loaded was placed upon the saturated 1,4-dioxane vapour atmosphere for over one day (Supplementary Fig. 18) to ensure that dynaCOF-330 crystals fully adsorbed 1,4-dioxane. All the sample preparation strategy is to ensure that 1,4-dioxane molecules are contained in the channels of dynaCOF-330 crystals while there is almost no 1,4-dioxane solvent outside crystals. Finally, the prepared TEM grid was quickly dropped into liquid nitrogen to stabilise 1,4-dioxane molecules in the channel of dynaCOF-330 crystals.

## Data availability

The data that support the findings of this study are available from the corresponding authors upon request. Refined crystal structures for dynaCOF-330 under vacuum, at stages III and IV upon acetone uptakes, and at stages I, III, and IV upon 1,4-dioxane uptakes have been deposited at the Cambridge Crystallographic Data Centre (CCDC#2226565–2226570) and provided as supplementary data. Source data are available. Source data are provided with this paper.

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

## Acknowledgements
The authors thank beamline-BL14B1 of the Shanghai Synchrotron Radiation Facility for providing the beamtime, T. Yang for assistance during synchrotron PXRD experiments, Dr N. Yu for assistance during PXRD data collection, Ms L. Long for assistance during gas/vapour adsorption measurements, and Dr R. Gao for assistance during data collection of fluorescent spectroscopies. We thank Prof. Omar M. Yaghi and the Berkeley Global Science Institute for the support of the scientific initiative. We thank Prof O. Terasaki and Prof K. D. M. Harris (Cardiff University) for the support of EM facilities (#EM02161943) at ChEM SPST, ShanghaiTech University and discussion on crystallography. We thank Prof W.-H. Zhu at East China University of Science and Technology, Prof M. Pan at Sun Yat-Sen University, Prof S. Duan at Fudan University, Dr Z.-Y. Shen at ZhenGe BioTech, Prof Z. Ning, Prof W. Liu, Prof Y. Huang, and Prof H. Liu at ShanghaiTech University for beneficial discussions. This work is supported by the Science and Technology Commission of Shanghai Municipality [Nos. 21XD1402300 (Y.B.Z), 21JC1401700 (Y.B.Z., J.S.), 21DZ2260400 (Y.B.Z, Y. M., Y.Z.), and 22QC1401500 (Y.Z)], the National Natural Science Foundation of China [Nos. 21522105 (Y.B.Z) 21875140 (Y.M.), and 22222108 (Y.M.)], the Double First-Class Initiative Fund of ShanghaiTech University (Y.B.Z.), and the China Postdoctoral Science Foundation No. 2020M681411 (L.W).

## Author contributions
Y.B.Z. initiated and led the research project. L.W. synthesised, characterised, and analysed all the materials. S.T. and Y.M. carried out the 3D ED experiments. Y.M. performed the PXRD Rietveld refinement. L.W., Z.S. and Y.Z. collected and analysed the fluorescent spectroscopy data. Z.X and S.J carried out the theoretical calculation. L.W and W.W. collected the PXRD data at SSRF. L.W., Y.Z and Y.B.Z. wrote the manuscript, and all the authors discussed and revised it together.

## Competing interests
The authors declare no competing interests.
