## [Peer Review File · Nature Communications]

Guest-adaptive molecular sensing in a dynamic 3D covalent organic frameworkReviewers' Comments:

Reviewer #1:

Remarks to the Author:

The authors submitted "Guest-adaptive molecular sensing in a dynamic 3D covalent organic framework" for publication in Nature communication. Even though the authors provided some interesting results in this manuscript, the novelty of this work is not high enough to publish in this journal. Here are my comments:

1. The TOC is missing.
2. The 3D monomer TAM is not new.
4. The authors did not provide any information about the porosity properties of this COF.
4. The chemical structure of COF should be investigated by solid-state NMR and XPS.
5. The authors should discuss the stability of COF in different conditions such as acidic and basic mediums.
6. the authors should provide different conditions such as different solvents to prepare this COF and study all properties of this COF including the porosity properties.

Reviewer #2:

Remarks to the Author:

Zhang et al. reported a 3D dynamic covalent organic framework which is flexible and can accommodate guest molecules, further recognizes environmentally significant gaseous analytes such as acetone, 1,4-dioxane and n-butane. This COF is a novel structure, which possesses good detecting performance. Moreover, the 3D COF is flexible, which is challenging. The mechanisms of sensing are clearly evidenced by single-crystal X-ray diffraction results. I recommend the manuscript be accepted after the following minor revisions.

1. Some important references of COFs should be cited. "Frontiers in Chemistry, 10, 943813, 2022", "Science China Chemistry, 10.1007/s11426-022-1350-1", Angew Chem Int Ed, 2017, 56 (51), 16313", "Chem Sci, 2013, 4 (12) , 4505-4511"
2. The Rwp and Rp of the Rietveld refinement in the main text should be added. The values should be moved from the SM to the main text.
3. The HRTEM images of the as-mentioned COFs should be provided in the main text.
4. An image to illustrate the DFT calculation process can be added to the main text.

Reviewer #3:

Remarks to the Author:

This is a very nice work. They have been able to fully characterize the dynamics of the COF under different VOCs and correlate their structures to their luminescent properties. The work has been thoroughly carried out and well presented. A very important work both from the perspective of basic chemistry and applied applications. Highly recommended for the publication in your prestigious journal.

Reviewer #4:

Remarks to the Author:

This is a really strong work, showing the introduction of anthracene units in a 3D-imine COF and its application for sensing of VOC. The material is characterized with a number of advanced techniques, in the pristine state and after adsorption of the various VOCs to give a profound insight into the dynamic structure change and adsorption mechanism of the respective gases. Thus, in my opinion this work deserves publication in Nature Communications.

There is indeed not much to criticize, just some smaller issues should be addressed.

1. Sensing of VOCs will be probably mainly important to be carried out in ambient air. However, this was not investigated. At least, fluorescent sensing of hydrocarbon gas was measured in a dry and humid nitrogen stream. The setup presented for this measurement should be also viable to measure fluorescent sensing of VOCs in air streams. Does oxygen influence the fluorescent response to the gas molecules?
2. Regarding the morphology of the materials. In Figure 2 c large crystals of dynaCOF-330 are shown. Do they stay intact after adsorption of gases/ after several cycling experiments or is there a change observed?
3. The yield of 62% for the dynaCOF-330 synthesis seems to be quite low. Is this due to a loss of material during filtration/purification or does the synthesis conditions yield no full conversion of monomers? Thus, what happens to the remaining 38% monomers?

Author Responses to Reviewer #1's Comments:

The authors submitted “Guest-adaptive molecular sensing in a dynamic 3D covalent organic framework” for publication in *Nature communication*. Even though the authors provided some interesting results in this manuscript, the novelty of this work is not high enough to publish in this journal.

Response: We appreciate the reviewer’s time in evaluating our work. We would like to highlight the novelty of this work lies in the clear demonstration of how concerted and adaptive structural transitions of dynaCOFs upon vapour adsorption would give rise to systematic changes in the fluorescence properties, which can be used for sensitive and reliable fluorescence sensing.

Here are my comments:

1. The TOC is missing.

Response: We found that the nature communication format does not have a TOC.

2. The 3D monomer TAM is not new.

Response: We agree with the reviewer that TAM monomer is not new. However, we merely used the TAM as a building block to construct a rationally designed dynamic COF incorporating fluorescent probes. We believe using TAM does not compromise the novelty of this work.

3. The authors did not provide any information about the porosity properties of this COF.

Response: We are sorry that the reviewer missed this information in the main text. We have characterized the porosity of the COF in the manuscript by CO₂ as probe molecule at 195 K and nitrogen at 77 K in Fig. S8, and organic vapours at 298K in Figs. 4a, 5a, S35, S36, S37. The porosity of DynaCOF-330 cannot be simply given by a surface area due to the dynamic nature of the framework, but it can be characterized by pore volume using these probe molecules.

4. The chemical structure of COF should be investigated by solid-state NMR and XPS.

Response: The solid-state NMR was in the manuscript (Fig S3). We have added the XPS data in the SM (Fig. S4). Chemical structure of COF is clearly characterized by ssNMR and FT-IR, shown the formation of imine bond accompanied by the disappearance of aldehyde and amino group.

On page 4: The presentation “The formation of imine bonds...solid-state nuclear magnetic response spectroscopy (ssNMR, Fig. S3).” was changed into “The formation of imine bonds...solid-state nuclear magnetic response spectroscopy (ssNMR, Fig. S3), and high resolution XPS (Fig. S4).”

Figure S4. High resolution X-ray photoelectron spectroscopy (XPS) of N 1s core level (a), O 1s core level (b) in dynaCOF-330. The N 1s signal peak at 399.1 eV assigned to the sp² nitrogen-carbon bond, and the O 1s signal peak at 533.4 eV assigned to sp³ oxygen-carbon bond.

5. The authors should discuss the stability of COF in different conditions such as acidic and basic mediums.

Response: We appreciate the reviewer for this suggestion. We did the chemical stability test after immersed COF for two hours in aqueous NaOH solution (pH = 14) and aqueous HOAc solution (pH = 2), as seen in Figs. S10-11.

On page 4: The presentation “DynaCOF-330 can stabilize up to 430 °C, characterized by thermal gravimetric analysis (Fig. S6)” was changed into “DynaCOF-330 can stabilize up to 430 °C, characterized by thermal gravimetric analysis (Fig. S7), and stabilize in alkali solution (Figs. S10-S11).”

Figure S10. PXRD patterns of dynaCOF-330 after treatment in aqueous NaOH solution (pH = 14, blue) and HOAc solution (pH = 2, red) for two hours. The COF is stable in alkali solution but lost long-range ordered structure in acid aqueous.

Figure S11. SEM images of COF-330 after treatment in various pH aqueous solutions.

6. the authors should provide different conditions such as different solvents to prepare this COF and study all properties of this COF including the porosity properties.

Response: The reported COF has been extensively optimized for its synthesis and activation. We believe this is routine COF synthesis condition screening and putting these data in the paper does not provide significant information for the authors. Here we summarized the PXRD pattern of the COF synthesized at several typical COF synthesis conditions for the reviewer's information.

Entry	Synthetic protocol	Solvent	Heating temperature (°C)	Modulator	Concentration of HOAc
1	vial	1,4-dioxane	80	aniline	6M
2	vial	1,4dioxane; toluene	80	aniline	6M
3	vial	1,4-dioxane; DCB	80	aniline	6M
4	tube	1,4-dioxane; DCB	120	no	6M
5	tube	1,4-dioxane; DMA	120	no	6M
6	tube	1,4-dioxane; n - butane	120	no	6M
7	tube	1,4-dioxane; toluene	120	no	6M
8	tube	1,4-dioxane	120	no	6M

Figure R1. PXRD patterns of dynaCOF-330 synthesized at the corresponding conditions.

Author Responses to Reviewer #2's Comments:

Zhang et al. reported a 3D dynamic covalent organic framework which is flexible and can accommodate guest molecules, further recognizes environmentally significant gaseous analytes such as acetone, 1,4-dioxane and n-butane. This COF is a novel structure, which possesses good detecting performance. Moreover, the 3D COF is flexible, which is challenging. The mechanisms of sensing are clearly evidenced by single-crystal X-ray diffraction results. I recommend the manuscript be accepted after the following minor revisions.

Response: We appreciate the reviewer's positive appraisal of our work.

1. Some important references of COFs should be cited. "Frontiers in Chemistry, 10, 943813, 2022", "Science China Chemistry, 10.1007/s11426-022-1350-1", Angew Chem Int Ed, 2017, 56 (51), 16313", "Chem Sci, 2013, 4 (12), 4505-4511"

Response: We have cited the excellent review on Fluorescent Covalent Organic Frameworks: A Promising Material Platform for Explosive Sensing. (reference 7)

On page 1: The presentation "Fluorescence sensing can be a viable approach to tackle this challenge,⁴⁻⁶" was changed into "Fluorescence sensing can be a viable approach to tackle this challenge,⁴⁻⁷"

2. The R_{wp} and R_p of the Rietveld refinement in the main text should be added. The values should be moved from the SM to the main text.

Response: We thank the reviewer for the suggestion, we have modified it accordingly.

(1). On page 4: The presentation "the high-resolution synchrotron PXRD pattern of the activated dynaCOF-330 (Fig. 2e) allows the Rietveld refinement against the structural model constructed based on the 3D ED results (Figs. S11&S13)." was changed into "the high-resolution synchrotron PXRD pattern of the activated dynaCOF-330 (Fig. 2e) allows the Rietveld refinement ($R_{wp} = 5.9\%$ and $R_p = 1.2\%$) against the structural model constructed based on the 3D ED results (Fig. 16)."

(2). On page 6: The presentation "The intermediate state has rhomboid channels, unit cell parameters of $a = 22.18 \text{ \AA}$, $b = 7.32 \text{ \AA}$, $c = 32.50 \text{ \AA}$, $\beta = 92.92^\circ$, and space group of $P2/n$ (Figs. 3e; Figs. S16; Table S1)." Was changed into "The intermediate state has rhomboid channels, unit cell parameters of $a = 22.18 \text{ \AA}$, $b = 7.32 \text{ \AA}$, $c = 32.50 \text{ \AA}$, $\beta = 92.92^\circ$, and space group of $P2/n$, which determined by PXRD Rietveld refinement with $R_{wp} = 4.5\%$ and $R_p = 1.6\%$ (Figs. 3e; Figs. S19; Table S1)."

(3). On page 6: The presentation "From Rietveld refinement analysis (Fig. S20)" was changed into "From Rietveld refinement analysis ($R_{wp} = 4.3\%$ and $R_p = 1.9\%$, Fig. S23)".

(4). On page 7: The presentation "the unit cell parameters changes from $a = b = 24.96 \text{ \AA}$; $c = 8.00 \text{ \AA}$ to $a = b = 27.84 \text{ \AA}$; $c = 7.26 \text{ \AA}$ (Figs. S21-S246)" was changed into "and the unit cell parameters changes from $a = b = 24.96 \text{ \AA}$; $c = 8.00 \text{ \AA}$ (obtained from Rietveld refinement with $R_{wp} = 7.5\%$ and $R_p = 5.2\%$, Fig S24) to $a = b = 27.84 \text{ \AA}$; $c = 7.26 \text{ \AA}$ ($R_{wp} = 7.8\%$ and $R_p = 5.4\%$, Fig. S25)"

3. The HRTEM images of the as-mentioned COFs should be provided in the main text.

Response: As the reviewer suggested, we have tried to acquire the HRTEM images of this COF, however, the sample is too thick to observe the lattice fringes.

Figure R2. HRTEM image of dynaCOF-330.

4. An image to illustrate the DFT calculation process can be added to the main text.

Response: We thank the reviewer for the suggestion, we have modified it accordingly.

(1). On page 20: The presentation of “Figure 4c, Inverse correlation of fluorescence emission intensity with 1,4-dioxane vapour uptakes with an inset for the steady 1,4-dioxane vapour sensing through adaptive inclusion.” was changed into “Figure 4e, The variation of dihedral changes of anthracene plane between selected time and the initial one, and the inset figure for the plane defined by the three carbon atoms marked by the orange circle.

(2). On page 6: The presentation of “The fluorescence sensing of 1,4-dioxane is reproducible by the vacuum-pressure swing of 1,4-dioxane pressure of 0.5 kPa and 3.3 kPa (inset, Fig. 4c).” was changed into “The fluorescence sensing of 1,4-dioxane is reproducible by the vacuum-pressure swing of 1,4-dioxane pressure of 0.5 kPa and 3.3 kPa (Figs. S33-S34).”

Author Responses to Reviewer#3's Comments:

This is a very nice work. They have been able to fully characterize the dynamics of the COF under different VOCs and correlate their structures to their luminescent properties. The work has been thoroughly carried out and well presented. A very important work both from the perspective of basic chemistry and applied applications. Highly recommended for the publication in your prestigious journal.

Response: We appreciate the reviewer's positive appraisal of our work.

Author Responses to Reviewer#4's Comments:

This is a really strong work, showing the introduction of anthracene units in a 3D-imine COF and its application for sensing of VOC. The material is characterized with a number of advanced techniques, in the pristine state and after adsorption of the various VOCs to give a profound insight into the dynamic structure change and adsorption mechanism of the respective gases. Thus, in my opinion this work deserves publication in *Nature Communications*.

Response: We appreciate the reviewer's positive appraisal of our work.

There is indeed not much to criticize, just some smaller issues should be addressed.

1. Sensing of VOCs will be probably mainly important to be carried out in ambient air. However, this was not investigated. At least, fluorescent sensing of hydrocarbon gas was measured in a dry and humid nitrogen stream. The setup presented for this measurement should be also viable to measure fluorescent sensing of VOCs in air streams. Does oxygen influence the fluorescent response to the gas molecules?

Response: According to the suggestion of the reviewer, we have provided the fluorescence response of this COF for nitrogen gas, oxygen gas, and carbon dioxide gas at 100 kPa and room temperature (Fig. S39). Exposing to 100 kPa oxygen gas, the fluorescence changes almost undetectable, quenched about 8%.

On page 8: The presentation of "Humidity often interferes with gas detection." was changed into "Humidity often interferes with gas detection, and we also examined other air components that they almost have no effect on fluorescence response (Fig S38)."

Figure S38. Fluorescence response of dynaCOF-330 for ambient air compositions and *n*-butane. Water vapour interferes the fluorescence response, and other air components that almost have no effect on fluorescence response.

2. Regarding the morphology of the materials. In Figure 2 c large crystals of dynaCOF-330 are shown. Do they stay intact after adsorption of gases/ after several cycling experiments or is there a change observed?

Response: According to the suggestion of the reviewer, we have provided the SEM images of COF-330 after ten cycles of 1,4-dioxane vapour adsorption (Fig. S34).

On page 6: The presentation of “The fluorescence sensing of 1,4-dioxane is reproducible by the vacuum-pressure swing of 1,4-dioxane pressure of 0.5 kPa and 3.3 kPa (inset, Fig. 4c).” was changed into “The fluorescence sensing of 1,4-dioxane is reproducible by the vacuum-pressure swing of 1,4-dioxane pressure of 0.5 kPa and 3.3 kPa (Figs. S32-S33).”

Figure S33. SEM images of COF-330 after ten cycles of 1,4-dioxane vapour adsorption.

3. The yield of 62% for the dynaCOF-330 synthesis seems to be quite low. Is this due to a loss of material during filtration/purification or does the synthesis conditions yield no full conversion of monomers? Thus, what happens to the remaining 38% monomers?

Response: To obtain well dispersed and homogeneous COF sample, aniline was added and would likely produced some soluble oligomers, therefore reducing the yield of COF.

Reviewers' Comments:

Reviewer #1:

Remarks to the Author:

I am satisfied with the changes made by the authors. The manuscript can now be published as it is.

Reviewer #2:

Remarks to the Author:

The authors solved all the referees' concerns in the revised manuscript. I suggest this manuscript to be accepted as its present form.

Reviewer #3:

Remarks to the Author:

The authors have addressed the reviewers' comments very well. It can be accepted now.

Reviewer #4:

Remarks to the Author:

The authors have answered all the concerns of this referee satisfactorily and therefore publication of this work can be recommended.

As a short note, one added sentence in manuscript and response-to-the-reviewers-letter is not similar and has a slightly different meaning, thus decide for the correct one.

Letter:

"Humidity often interferes with gas detection, and we also examined other air components that they almost have no effect
no (on?) fluorescence response"

Manuscript:

"Humidity often interferes with gas detection, and we also examined other air components that they almost have no fluorescence response"

Author Responses to Reviewer #1's Comments:

I am satisfied with the changes made by the authors. The manuscript can now be published as it is.

Response: We appreciate the reviewer's positive appraisal of our revised manuscript.

Author Responses to Reviewer #2's Comments:

The authors solved all the referees' concerns in the revised manuscript. I suggest this manuscript to be accepted as its present form.

Response: We appreciate the reviewer's positive appraisal of our revised manuscript.

Author Responses to Reviewer #3's Comments:

The authors have addressed the reviewers' comments very well. It can be accepted now.

Response: We appreciate the reviewer's positive appraisal our revised manuscript.

Author Responses to Reviewer #4's Comments:

The authors have answered all the concerns of this referee satisfactorily and therefore publication of this work can be recommended.

Response: We appreciate the reviewer's positive appraisal of our revised manuscript.

As a short note, one added sentence in manuscript and response-to-the-reviewers-letter is not similar and has a slightly different meaning, thus decide for the correct one.

Letter:

"Humidity often interferes with gas detection, and we also examined other air components that they almost have no effect no (on?) fluorescence response"

Manuscript:

"Humidity often interferes with gas detection, and we also examined other air components that they almost have no fluorescence response"

Response: We are sorry for the typo. The sentence now reads: "Humidity often interferes with gas detection. We also examined other air components and found that they almost have no fluorescence response".